# Federated Learning Approach for Early Detection of Chest Lesion Caused by COVID-19 Infection Using Particle Swarm Optimization

Dasaradharami Reddy Kandati and Thippa Reddy Gadekallu *

School of Information Technology and Engineering, Vellore Institute of Technology, Vellore 632014, India
* Correspondence: thippareddy.g@vit.ac.in

**Abstract:** The chest lesion caused by COVID-19 infection pandemic is threatening the lives and well-being of people all over the world. Artificial intelligence (AI)-based strategies are efficient methods for helping radiologists by assessing the vast number of chest X-ray images, which may play a significant role in simplifying and improving the diagnosis of chest lesion caused by COVID-19 infection. Machine learning (ML) and deep learning (DL) are such AI strategies that have helped researchers predict chest lesion caused by COVID-19 infection cases. But ML and DL strategies face challenges like transmission delays, a lack of computing power, communication delays, and privacy concerns. Federated Learning (FL) is a new development in ML that makes it easier to collect, process, and analyze large amounts of multidimensional data. This could help solve the challenges that have been identified in ML and DL. However, FL algorithms send and receive large amounts of weights from client-side trained models, resulting in significant communication overhead. To address this problem, we offer a unified framework combining FL and a particle swarm optimization algorithm (PSO) to speed up the government's response time to chest lesion caused by COVID-19 infection outbreaks. The Federated Particle Swarm Optimization approach is tested on a multidimensional chest lesion caused by the COVID-19 infection image dataset and the chest X-ray (pneumonia) dataset from Kaggle's repository. Our research shows that the proposed model works better when there is an uneven amount of data, has lower communication costs, and is therefore more efficient from a network's point of view. The results of the proposed approach were validated; 96.15% prediction accuracy was achieved for chest lesions caused by the COVID-19 infection dataset, and 96.55% prediction accuracy was achieved for the chest X-ray (pneumonia) dataset. These results can be used to develop a progressive approach for the early detection of chest lesion caused by COVID-19 infection.

**Keywords:** chest lesion caused by COVID-19 detection; machine learning; Federated Learning; image processing; particle swarm optimization

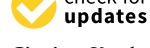



## 1. Introduction

Chest lesion caused by COVID-19 infection has impacted human health and life in a significant way due to its rapid spread. Researchers have found that the chest lesion caused by COVID-19 pandemic is one of the most catastrophic health problems now impacting millions of individuals throughout the world [1–3]. A person infected with COVID-19 releases respiratory particles into the air when he or she sneezes, coughs, or talks. Detection of chest lesion caused by COVID-19 infection at the earliest possible stage is crucial to limiting the potential spread of the disease. There is a growing need for a quick and accurate technique for diagnosing chest lesion caused by COVID-19 infection outbreaks. Chest lesion caused by COVID-19 infection was declared a public health emergency by the World Health Organization (WHO) due to its outbreak severity. This deadly outbreak caused many people to suffer in many ways. Globally, millions of tests have been performed

to detect chest lesion caused by COVID-19 virus [4]. The detection of chest lesion caused by COVID-19 infection in the human body currently uses one of these three methods:

- The COVID can be detected using computed tomography scans (CT scans), which use 3D radiographic images.
- Contagious RNA can be detected from nasal swabs using Reverse Transcription Polymerase Chain Reaction (RT-PCR).
- Less equipment is required for chest X-rays (CXR) and these are more portable than CT scan machines. The CXR test also takes about 15 s per person, which is time efficient.

However, hospitals and many health centres does not have the facility for CT scans. Also, many hospitals do not have an easy access to the equipment to diagnose chest lesion caused by COVID-19 infection using RT-PCR test. This is also a time consuming process. In this paper we use chest lesion caused by COVID-19 infection chest X-ray image dataset because of its ease of availability, reliability and accuracy.

One of the most common types of clinical testing is RT-PCR, which uses a swab (inserted into the mouth or nose) to collect a sample for pathogen RNA sequencing from viral specimens. The purpose of collecting these antibodies is to determine if the disease has spread throughout the body. Another common strategy is to explore RNA sequences since this helps to identify antibodies responsible for preventing viruses and necessitates the use of FDA-approved medicines to combat them. These clinical trials proved to be beneficial. However, these diagnostic treatments require the help of doctors and take a long time. Because of this, they are quite expensive.

The earliest possible diagnosis of a chest lesion caused by COVID-19 infection may be made if medical professionals have access to sufficient epidemiological data, including protein composition (mostly RNA), serological findings, and diagnostic investigations. Moreover, the clinical importance of a chest lesion caused by COVID-19 infection can be determined with high precision using imaging data [5]. Early diagnosis of a chest lesion caused by COVID-19 infection in clinical specimens is a drawback for health care and epidemic management. The spread of disease can be controlled by early detection [6,7].

Artificial Intelligence (AI) engineers and data scientists are in a great position to detect the spread of a chest lesion caused by the COVID-19 infection. The researchers have started implementing Machine Learning (ML) techniques, particularly Deep learning (DL), on various chest X-ray images. Despite this, medical organisations, which often manage CXR images, permit using their data for model training. Medical institutions with fewer samples cannot train a model that can detect chest diseases with the expected performance because medical data are subject to stringent privacy regulations [8–10]. Computer-aided diagnosis has advanced considerably in many healthcare domains with recent developments in AI [11]. DL techniques were used in the past which use the Convolutional Neural Network (CNN) classification model to find diseases like cancer and pneumonia. Chest X-ray images can be used to predict chest lesions caused by COVID-19 infection based on the DL models' promising results [12,13]. AI and DL models can be used to predict how the disease will spread based on previous occurrences. This could retain the diseases from spreading further [14]. Thus, there is a requirement to develop ML models to detect chest lesions caused by COVID-19 infected patients or to predict their future disease transmission. However, this is challenging, since patient data is sensitive, and without sufficient data, it is impossible to develop an accurate model [11]. Thus, there is a need to develop a model that makes accurate predictions without requiring the transfer of a patient's personal data.

Google first presented Federatyed Learning (FL) as a new paradigm for ML in 2016. The use of FL allows us to potentially build a ML model from different datasets without revealing any private information [13,15–17]. Furthermore, FL guarantees a decrease in communication costs between the server and the client [18]. Since the client-side data is not sent to the server for training, the delay in sending and receiving information is minimized. In addition to use of large volumes of data locally, FL can do the same thing remotely [19]. But with FL, communication is more time-consuming than computation. In order to make FL more effective the amount of network communication time needs to be cut down. In

recent studies [20–24], FL can be used to accurately diagnose a chest lesion caused by COVID-19 infection using X-ray images. However, the aforementioned research utilised FL's default configuration, which leads to poor effectiveness when client data uncertainty is present and necessitates a substantial computational cost for the distribution of model updates. Particle Swarm Optimization (PSO), a method that finds the best solution in a decentralized setting, is used in the proposed research to speed up model updates [25,26]. PSO needs many repetitions because it uses a random method to find the best solution. The PSO works effectively in adaptive and complex settings, such as FL. Thus, we create an innovative method where the PSO is used in FL.

*Key Contributions*

The major contributions of this paper are summarized below:

1. We develop a strategy for addressing communication delay. Our approach proposes a new paradigm of efficient comprehensive integration of FL and PSO, therefore applying the well acknowledged benefits of swarm intelligence to distributed learning applications.
2. The empirical assessments demonstrate that the suggested method is more effective to the standard FL methods in terms of performance. The suggested FPS Optimization displays an improvement in terms of its accuracy.
3. Based on the results of the simulation studies, it is evident our method is better than the benchmark processes in terms of achieving higher levels of accuracy during testing.

The remaining sections of the paper are organized as follows. The Section 2 discusses earlier research that makes use of FL and PSO. Section 3 provides an overview of the proposed methodology. Section 4 presents an evaluation of the proposed technique. Finally, the paper is concluded with Section 5.

## 2. Background and Related Work

This section reviews the current literature on FL, PSO, and chest lesions caused by COVID-19 infection detection and analysis.

### 2.1. Federated Learning

FL is an effective strategy for learning from dispersed datasets [19,27]. It prevents sensitive information from escaping when training a model on data from several devices. Recently, there has been a lot of focus on FL, which has motivated several useful initiatives to create learning-based applications on a wide variety of decentralized devices. FL enables decentralized learning without the need to transfer raw data between nodes, thereby protecting user information [27,28]. Moreover, FL guarantees a decrease in the server-client communication cost [18]. The communication between the client and server has decreased since training-related client data is not transferred to the server. The benefits of FL include increased privacy and decreased communication costs. FL is used in situations where maintaining confidentiality and privacy are of the utmost importance [1,28,29]. Figure 1 explains the FL process.

1. The server generates a model using the given data.
2. Sends a copy of the model to each client, who will then train the model based on their local data.
3. Client-trained models are uploaded to the server for further processing. Please keep in mind that only the model is being used, not any actual data.
4. Aggregation algorithms are used on the server side to add up the models sent by each client.
5. The server sends updates to the clients, and the cycle keeps going until the model is optimized.

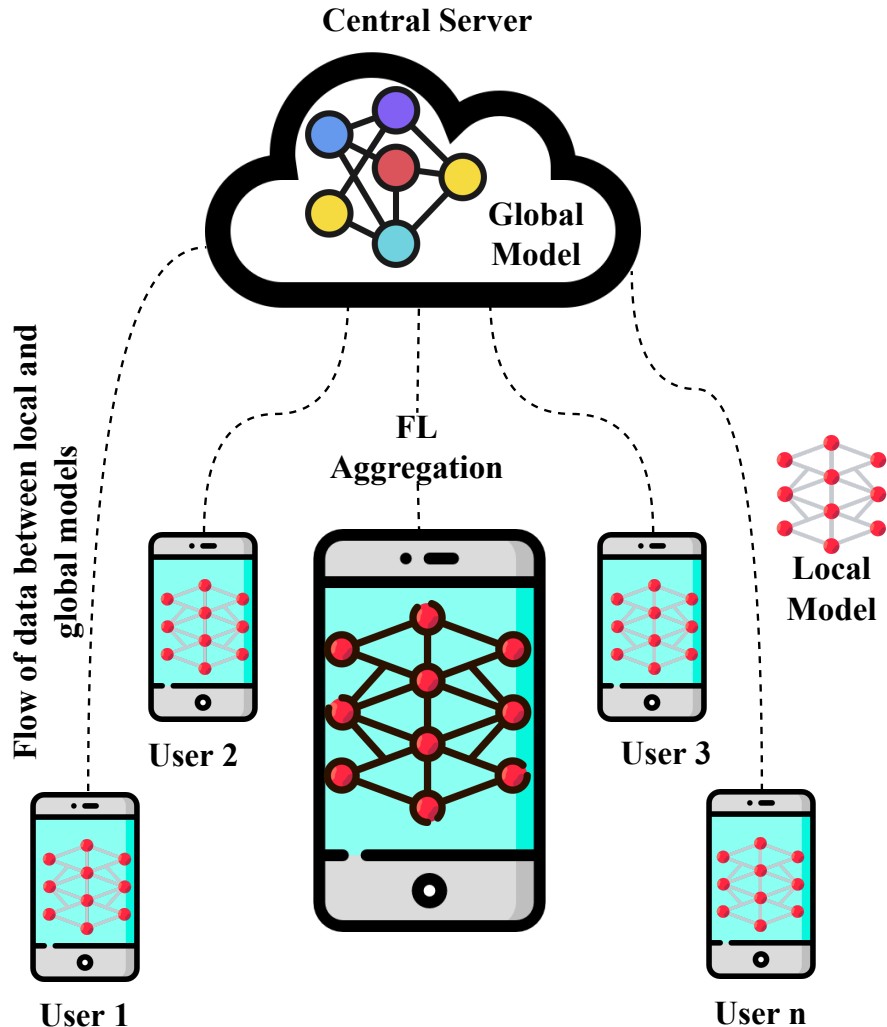

**Figure 1.** Federated Learning Architecture.

There are many FL services that use strategies such as Fedavg, Federated Stochastic Gradient Descent (FedSGD) [18], and FedMA [30] to carry out Step 4. The algorithm was originally developed by McMahan [18] and is currently used to enhance models obtained from collected data in several FL studies. Global models are generated by averaging data from all clients in both algorithms. FedSGD obtains the weights on the server instead of updating them on the client. Using a gradient, this algorithm figures out the average, and then it changes the global weights to make a global model. By combining FedSGD and mini-batches, the FedAvg algorithm updates models directly on the client. The server then takes the average of these weights to make a new global model.

### 2.2. Particle Swarm Optimization

Kennedy and Eberhart created the PSO in 1995 [25,26], based on the behavoir of birds. The method uses techniques inspired by natural bird and fish swarms to optimise several variables at once. The PSO algorithm saves time and memory because it is easy to use, converges quickly, is strong, can be scaled up, and works well with mathematical problems [31]. Stochastic optimization uses a statistical technique and needs several iterations.

The components of PSO consist of swarms and particles. A swarm is a collection of several different particles. Particles can be viewed as symbols for the infinite number of potential outcomes of a problem. The next step of each particle is determined by its

position and speed (S). To determine the ideal value for the entire structure, every particle communicates with its peers to provide its *pb* (particle best) variable. The *gb* (global best) parameter for every particle is set to the value that maximizes the *pb* parameters: $gb = max_i(pb)$. Based on a particle's present position, its distance from its past position, its current velocity, and its current distance from its past position, the particle's position is modified. There is only one objective function required, which makes it different from other optimization algorithms. Iterative optimization is achieved using the PSO method. Using the Equation (1) below, particles calculate the speed of their next step by adding their inertia ($S^{t-1}$, the speed of their last step), and their *pb* and *gb* values.

$$S_i^t = \alpha \cdot S_i^{t-1} + c_1 \cdot rand_1 \cdot (pb - Si^{t-1}) + c_2 \cdot rand_2 \cdot (gb - S_i^{t-1}) \tag{1}$$

In Equation (1), $\alpha$ represents the *inertia* weight, $c_1$ and $c_2$ are the acceleration constants, called the cognitive and social parameters respectively, and $rand_1$ and $rand_2$ are two random numbers in the range [0, 1] that restrict the velocity of the particle in the coordinate direction. The randomization provided by the parameters $c_1$, $c_2$, $rand_1$, and $rand_2$, reduces the predictability of the method while increasing its adaptability. Here, *pb* is the particle best, and *gb* is the global best.

*2.3. Related Work*

Several research findings have examined the effectiveness of FL by improving communication between clients. Many problems are caused by the unstable network environment of mobile devices in FL. There are many problems with this, such as frequent crashes, frequent changes in node groups, a lot of work for the central server, and a big increase in latency as the number of nodes increases. In some cases, it is also important to consider the volume of information being sent between the client and server. The recent studies in [32–34] propose a strategy for tackling the limited bandwidth bottleneck by jointly selecting devices and designing beamforming algorithms.

PSO simulates animal swarm intelligence to address challenging optimization problems without convexity or differentiability [25,35]. PSO concepts have been used in some recent attempts to enhance ML performance. PSO is used to improve the accuracy of recognition and image classification with convolutional neural networks (CNNs) [36–38]. FL performance can be improved through the integration of FL and PSO [39,40]. The FL is used for learning [39], whereas the PSO is used for finding the most optimal hyper-parameters. There have been a lot of works on how to improve FL performance. Most of the works highlighted about the client communication and global optimization. In addition, a variety of approaches have been proposed for implementing PSO in FL, but PSO has not been used to improve the network communication between global models.

PSO is generally used to select an appropriate client model for every round of global model updates. The PSO algorithm optimizes the performance of FL by fine-tuning its parameters. During aggregation, PSO optimizes the coefficients of clients involved in the process [41]. The integration of FL and PSO allows us to identify the function that a group of agents must optimize [42]. The authors in [43] combined PSO and FL to meet both privacy and feature selection requirements. The authors in [40] proposed a FedPSO technique that is based on particle swarm optimization in order to increase the network communication performance of FL and to minimize the quantity of the data that is transmitted from clients to servers. In another work, the authors in [44] propose a comprehensive method that is based on FL with a particle swarm optimization algorithm that allows investigators to respond to forest fires more quickly. To improve FL's efficiency, most previous works focused on expanding communication with clients and performing global optimization. However, PSO has been used to optimize local optimization problems; it has never been applied to boost the efficiency of global models through the transmission of data over the internet. Our work is mostly about making FL run better by modifying the type of information used when clients and servers exchange data based on PSO. Table 1 summarizes the key findings from the above discussion.

**Table 1.** Summary of Important Surveys on FL and PSO Algorithms.

| Ref. No | Technologies Used | Key Contributions | Limitations |
|---|---|---|---|
| [45] | Generative adversarial network (GAN) model. | TD-GAN synthesis and segmentation without X-ray images are demonstrated using a deep architecture. | Labeling X-ray images is hard because parts of anatomy overlap and the texture patterns are complicated. |
| [46] | CheXNet algorithm. | CheXNet can detect pneumonia in frontal chest X-rays more accurately than human radiologists. | The model and radiologists were not permitted to use the patient's history, which complicates the radiologists' diagnoses. |
| [47] | Hierarchical convolutional neural network method. | The loss function was changed to work better, and a more compact model structure was developed. | In medical image processing, the use of DL networks has caused overfitting and a lack of transfer efficiency. |
| [48] | InstaCovNet-19 model. | Several preprocessing and training strategies were used to increase classification accuracy. | ML based models struggle most due to insufficient feature extraction and poor preprocessing of input images. |
| [40] | FedPSO | Using PSO to optimize FL communications can help minimize network expenses significantly. | Current FL clustering methods broadcast and receive many weights, which reduces their accuracy in unstable networks. |
| [41] | FL and PSO | During the aggregation, PSO optimizes the weight of each client. | It is possible to improve FL models further since they are not stable. |
| [42] | PSO + FL = PAASO | Agents can optimize functions that need to be understood. | In heterogeneous environments, it shows poor performance. |
| [44] | PSO and FL | This centralised approach will benefit the early detection and diagnosis of forest fires. | The algorithm does not analyze other nature-inspired algorithms. |
| **This paper** | **FPS Optimization** | • **FPS Optimization works better than traditional FL methods like FedAvg, and it takes into account how settings and other factors affect learning for remote workers.**<br>• **Early detection of chest lesions caused by COVID-19 infection and pneumonia will be easier with this unified framework.** | **Traditional FL takes more time to tune the parameters.** |

From the above literature survey, it is clear that PSO is capable of addressing FL challenges. In order to prevent the spread of chest lesions caused by COVID-19 infection, we have utilized FPS Optimization to train local data models based on ML at each individual location, then transmit those models to a centralized server where they are aggregated to form a global data model. A CNN classifier was trained on a server data model to make chest lesions caused by COVID-19 infection predictions. To the best of our knowledge, this is the first study to propose the FPS Optimization framework for early prediction of COVID-19-induced chest lesions.

Our work outperformed the previous works performed on chest lesions caused by the COVID-19 infection dataset. Since the FL technique used here, the client privacy is ensured. The proposed model also uses PSO along with FL which optimizes the client selection strategy and thereby enables the better and faster performance. Our work is superior in implementing both the smaller chest lesion caused by COVID-19 images and the larger chest X-ray images (pneumonia) datasets.

### 3. Proposed Methodology

The main goal of our study is to come up with a way to detect chest lesions caused by COVID-19 infection as early as possible and stop the loss that occurs due to it. Such a model is possible by combining FL and PSO. To enhance the accuracy of CNN models, it is common procedure to add more layers to the model. This is known as a deep neural network. The training effort required for the weight parameters grows proportionally with the layer depth. The cost of sending the model learned on the client to the server increases, as shown in Figure 2. Thus, we propose a unified framework, which uses PSO features to transfer the trained model, regardless of its size, with the highest score (for example, accuracy or loss).

$$W_{t+1}{}^{avg} \leftarrow \sum{}^{K}{}_{K=1} W_t{}^{k}/K$$

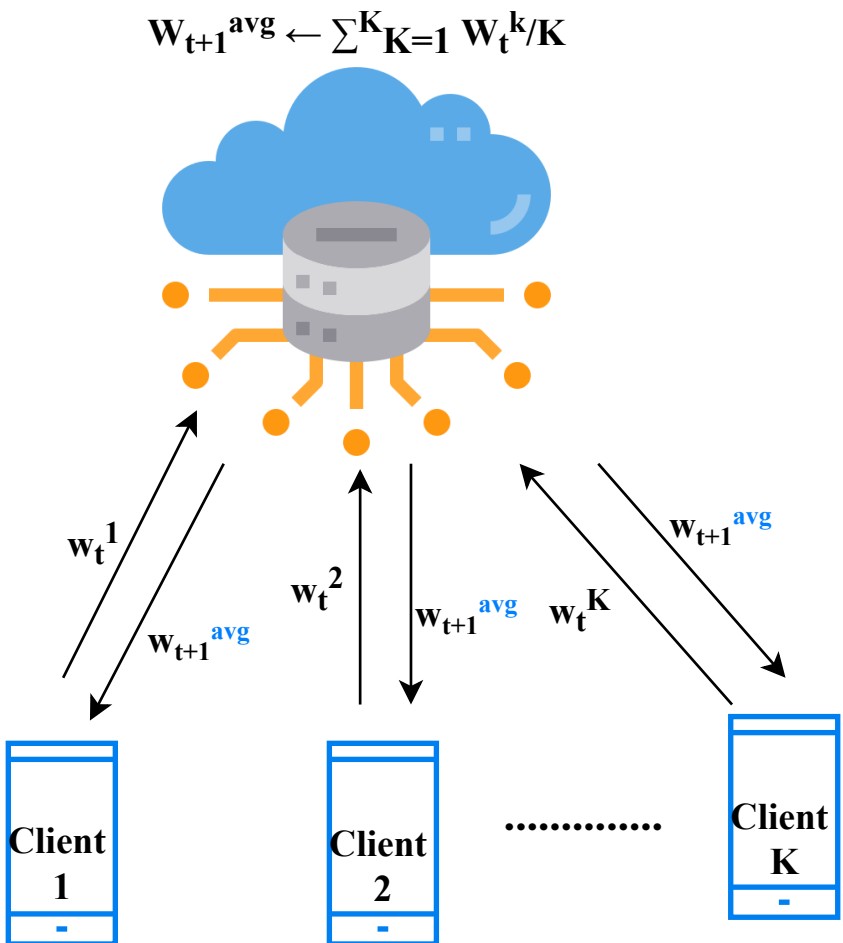

**Figure 2.** Federated Averaging Process. FL obtains the average of the $w_t$ value received from the client of K from the server and sends the updated $w_{t+1}$ back to the client.

First, we will analyze the algorithm that was utilized in the work done before on FL (such as FedAvg [18]), and then we will discuss the suggested FPS Optimization. The steps involved in the first process of Algorithm 1, which is employed in FL, are as follows: line 4 indicates which client will participate in the round. In lines 5 and 6, the client's weight values are received. The average weights are calculated from line 7, and then the global weights are determined. Lines 8–10 show how the client gets the information based on the global weights sent by the server.

---

**Algorithm 1** FedAvg algorithm; K = no. of clients; E = Epochs served for the whole client; Clients can be chosen based on their C-ratio.

---

1: **function** SERVERAGGREGATION($\eta_N$)
2:     initialise $w_0$
3:     **for** every iteration $t = 1, 2, \ldots$ **do**
4:         $S_t \leftarrow$ (clients are chosen randomly from set of $max(CK, 1)$)
5:         **for** each client $kES_t$ in parallel **do**
6:             $w_{t+1}^k \leftarrow$ UPDATECLIENT($k, w_t$)
7:         $w_{t+1} \leftarrow$ average of the weights that are collected $w_{t+1}^k$ of $S_t$ clients
8: **function** UPDATECLIENT($k, w$)
9:     Carry out the process of learning on the client $k$ with weight $w$ till the client arrives $E$ epoch
10:     $w \leftarrow$ (revised weight after receiving new information)
11:     **return** $w$ to the server

---

Next, the suggested model, FPS Optimization, accepts model weights exclusively from the client with the highest score and not from all clients. Figure 3 shows how the process works. The highest score is determined by lowest possible loss value of the trained client. The loss value in this case is only 4 bytes long. FPS Optimization finds the finest model by applying the *pb* and *gb* variables, and it then changes the value of *S* for each weighted array member that represents the finest model.

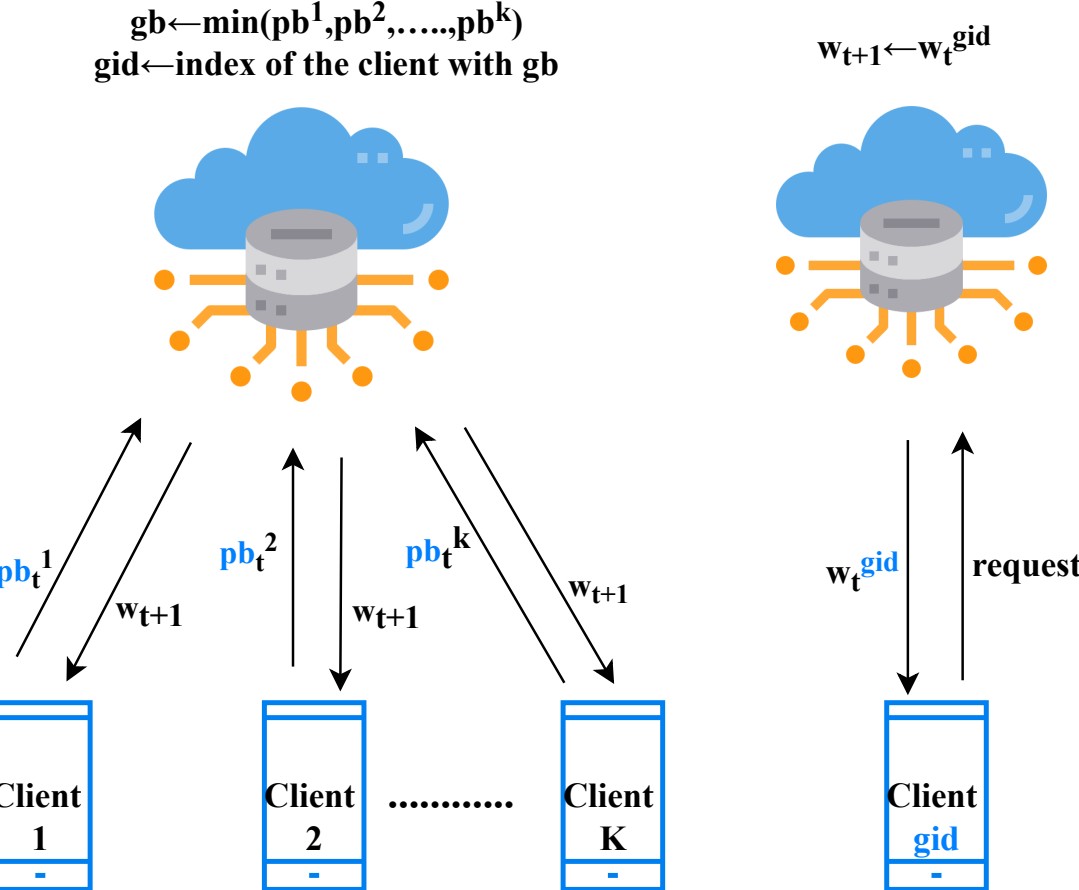

**Figure 3.** FPS Optimization Process. The server receives a client's score and requests a learning model from the client who submits the optimal value to set it as a global model.

According to Equation (1), the FPS Optimization weights were updated as follows:

$$S_l^t = \alpha \cdot S_l^{t-1} + c_1 \cdot rand_1 \cdot (pb - Sl^{t-1}) + c_2 \cdot rand_2 \cdot (gb - S_l^{t-1})$$
$$w_i^t = w_i^{t-1} + S^t \tag{2}$$

The weight w of each layer in CNN is represented by $S$ in Equation (2). The present step weight $w^t$ is calculated by adding $S$ to the weight $w^{t-1}$ from the preceding step. As in Equation (1), $\alpha$ represents the inertia weight, $c_1$ denotes the constant of acceleration for $pb$, and $c_2$ denotes the constant of acceleration for $gb$. $Rand_1$ and $Rand_2$ are randomly generated numbers between 0 and 1.

The concept of the proposed framework is based on Algorithm 2. This algorithm extends Algorithm 1 by applying PSO. In contrast with traditional methods, the function ServerAggregation only accepts $pb$ values from the client on line 5. In ServerAggregation function the variables $w_0$, $gid$, $pb$, $gb$ are initialized. Here the UpdateClient function is called, which updates the $pb$ value. The $w^{gb}$ and $pb$ are compared and $w^{gb}$ is updated with the new $pb$ value and $gid$ is updated with K value which is client id. Using lines 6–8, we search for the client with the $pb$ value (lowest loss value after training the client) among the available samples. Using the PSO, CNN performs the UpdateClient function. In UpdateClient function the variables $S$, $w$, $w^{pb}$, $\alpha$, $c_1$, $c_2$ are initialized. $\beta$ is a constant that holds the divided batch sizes. For each layer the velocity of the particle is updated. Simultaneously, the weights are also updated based on the batch sizes. Variable S is calculated in lines 13–14 along with the best value of $w^{pb}$, which is stored by the user, and the $w^{gb}$ value received from the server. For each layer weight, the process is repeated. Line 15 adds Variable S to the previous round's $w$ to figure out the current round's $w$. Continue training through lines 16–18 until client epoch E has been reached. GetAptModel requests the best model from the client on the server (lines 20–23). The GetAptModel will make a request to the client with $gid$ and gets $w$ as acknowledgement from client.

---

**Algorithm 2** Federated Particle Swarm Optimization Algorithm

---

1: **function** SERVERAGGREGATION($\eta_N$)
2:     initialize $w_0$, $gid$, $pb$, $gb$,
3:     **for** every iteration $t = 1, 2, \ldots$ **do**
4:         **for** every parallel client $k$ **do**
5:             $pb \leftarrow$ UPDATECLIENT($k, w_t^{gid}$)
6:             **if** $gb > pb$ **then**
7:                 $gb \leftarrow pb$
8:                 $gid \leftarrow k$
9:         $w_{t+1} \leftarrow$ GETAPTMODEL($gid$)
10: **function** UPDATECLIENT($k, w_t^{gid}$)
11:     initialize $S$, $w$, $w^{pb}$, $\alpha$, $c_1$, $c_2$
12:     $\beta \leftarrow$ (divide $p_k$ into batches each of size $B$)
13:     **for** every layer with weight $l = 1, 2, \ldots$ **do**
14:         $S_l \leftarrow \alpha \cdot S_l + c_1 \cdot rand \cdot (w^{pb} - S_l) + c_2 \cdot rand \cdot (w_t^{gb} - S_l)$
15:     $w \leftarrow w + S$
16:     **for** every epoch of client i from 1 to $E$ **do**
17:         **for** batch $b \in B$ **do**
18:             $w \leftarrow w - \eta\delta(w; b)$
19:     **return** pb to the server
20: **function** GETAPTMODEL($gid$)
21:     make a request to Client($gid$)
22:     will be acknowledged $w$ from Client
23:     **return** $w$ to the server

---

## 4. Experiments and Results

This section summarizes the experiments conducted to evaluate the FPS Optimization architecture. The experiment was conducted using the chest lesion caused by COVID-19 infection dataset and the chest X-ray (pneumonia) dataset, as discussed in the next subsection. Experimental results for accuracy are discussed in the subsection on test results for accuracy for the chest lesion caused by the COVID-19 infection dataset and the chest X-ray (pneumonia) dataset. In the last subsection, we'll compare how well FPS optimization and regular FL work on the chest lesion caused by the COVID-19 infection dataset and the chest X-ray (pneumonia) dataset.

### 4.1. Experimental Setup

The experiments were carried out on a laptop equipped with an Intel(R) Core(TM) i5-1135G7 @ 2.40GHz CPU, two NVIDIA GeForce RTX 2070 Super GPUs each with 8 GB of RAM, and 64 GB of memory. We used Keras version 2.4.3 and TensorFlow version 2.3.0 to write our experimental code. The purpose of the work was to enhance FL's communication capabilities. The weights are updated by a PSO algorithm between the client and the server.

The experiment was conducted using the chest lesion caused by the COVID-19 infection dataset. This set of images consists of $150 \times 150$ pixels, divided into three categories: Covid, Normal, and Viral Pneumonia, and it has both training images and test images. We assigned particle numbers to each particle and shuffled the datasets for training. Both FL and FPS Optimization use Adam Optimizer to train the client rate. The learning rate value was 0.020. Table 2 depicts the hyper-parameter values used in the paper.

**Table 2.** Constants.

|              | FedAvg             | FPS Optimization |
| ------------ | ------------------ | ---------------- |
| Client       | 10                 | 10               |
| C-value      | 0.1, 0.2, 0.5, 1.0 | -                |
| Epoch        | 30                 | 30               |
| Client-Epoch | 5                  | 5                |
| Batch        | 10                 | 10               |

### 4.2. Dataset

The dataset used in this work is taken from the kaggle repository https://www.kaggle.com/datasets/pranavraikokte/covid19-image-dataset (accessed on 5 September 2022).

The dataset totally contains 317 images. The training dataset folder has 251 images and testing dataset folder has 66 images.

The images in each folder belong to three categories namely:

- Covid
- Normal
- Viral Pneumonia

The purpose of this study is to use PSO to detect the onset of a COVID-19-induced chest lesion. Figure 4 indicates the sample images from the chest lesion caused by the COVID-19 infection dataset.

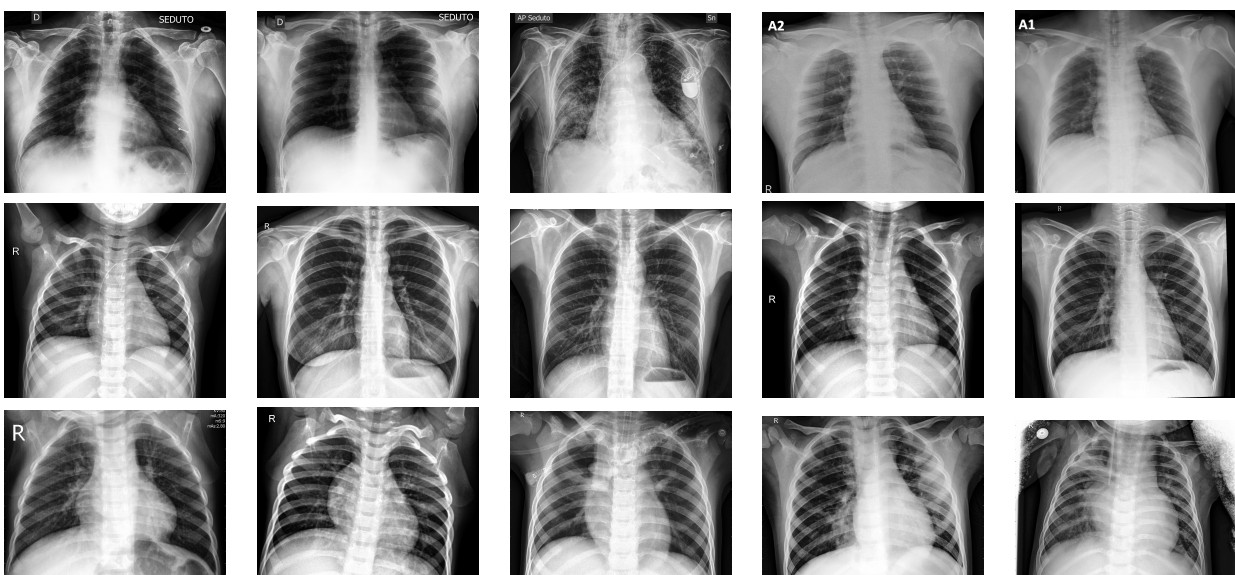

**Figure 4.** Sample images from the chest lesion caused by the COVID-19 infection dataset. The images are divided into three categories: Covid, Normal, and Viral Pneumonia.

The chest X-ray (pneumonia) dataset also used in this work is taken from the Kaggle repository https://www.kaggle.com/datasets/paultimothymooney/chest-xray-pneumonia (accessed on 15 January 2023 ). The dataset totally contains 5856 images. The dataset is organized into 3 folders (train, test, val). The train dataset folder has 5216 images, test dataset folder has 624 images and val dataset folder has 16 images. The images in each folder belong to two categories namely:

- NORMAL
- PNEUMONIA

Figure 5 indicates the sample images from the chest X-ray images (pneumonia) dataset.

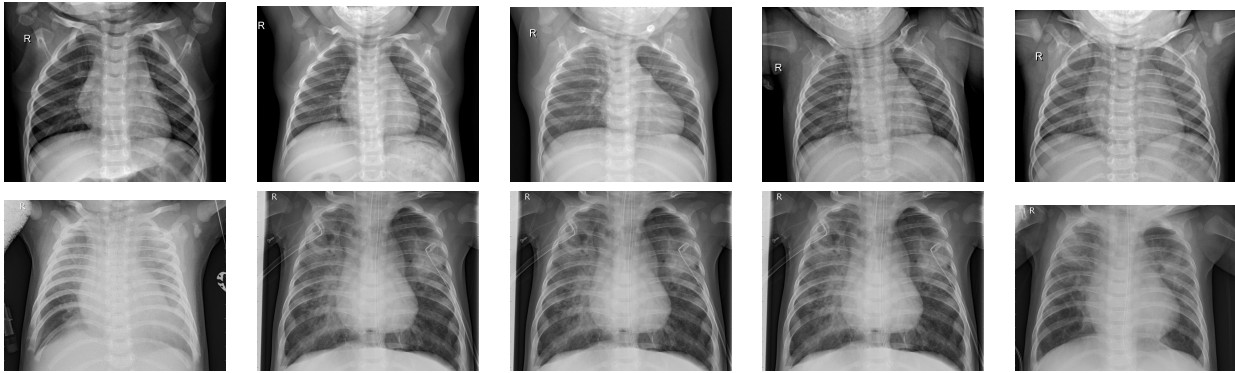

**Figure 5.** Sample images from the chest X-ray (pneumonia) dataset. The images are divided into two categories: Normal, and Pneumonia.

### 4.3. Dataset Partitioning

The chest lesion caused by the COVID-19 infection dataset is split into three versions: one with 137 images called "Covid," one with 90 images called "Normal," and another with 90 images called "Viral Pneumonia". Chest X-rays are structured into the testing and training directories. We used 80% of the data for training and 20% for testing. Table 3 shows the data partitioning details.

**Table 3.** Dataset Division.

| Dataset | Training | Testing | Total |
| --- | --- | --- | --- |
| Covid | 111 | 26 | 137 |
| Normal | 70 | 20 | 90 |
| Viral Pneumonia | 70 | 20 | 90 |
| Total | 251 | 66 | 317 |

The chest X-ray (pneumonia) dataset is split into two versions: one with 1583 images called "Normal", and another with 4273 images called "Pneumonia". Chest X-rays are structured into the testing, training, and validation directories. We used 80% of the data for training, 10% for testing, and 10% for validation. Table 4 shows the data partitioning details.

**Table 4.** Dataset Division for Chest X-ray (Pneumonia).

| Dataset | Training | Testing | Validation | Total |
| --- | --- | --- | --- | --- |
| Normal | 1341 | 234 | 8 | 1583 |
| Pneumonia | 3875 | 390 | 8 | 4273 |
| Total | 5216 | 624 | 16 | 5856 |

*4.4. CNN*

The CNN is a method for obtaining different classification models by backpropagation neural networks. ML methods that utilize DL include multi-level nonlinear transformations. The most popular CNN at the moment is a deep neural network. CNN takes their connection pattern from the structure of the visual cortex of animals, which is one of the most widely used structures in neuroscience. The characteristics of weight sharing, local connection, and pooling in CNN minimize network complexity, model invariance, and training parameters for distortion, translation, and scaling. Along with being reliable and able to handle mistakes, the programme is also easy to train and optimize. These superior properties make it superior to fully connected neural networks in a number of different applications.

In our work, we used a three-layer CNN model that is used to conduct the experiments (the first layer with 32 channels, the second layer with 64, and the third layer with 128 channels, each followed by 2 × 2 max pooling). Table 5 shows the layers of the corresponding model.

**Table 5.** Parameters settings for the CNN.

| Layer | Shape |
| --- | --- |
| Layer 1 | Conv2D (32, 3, 3) ReLU, MaxPool2D (2, 2) |
| Layer 2 | Conv2D (64, 3, 3) ReLU, MaxPool2D (2, 2) |
| Layer 3 | Conv2D (128, 3, 3) ReLU, MaxPool2D (2, 2) |
| Layer 4 | Dense (512) ReLU |
| Layer 5 | Dense (2) Softmax |

*4.5. Test Results for Accuracy*

Figure 6 and Table 6 shows the accuracy test results for the chest lesion caused by the COVID-19 infection dataset. Test accuracy determines the design of all of these graphs. FPS Optimization provided higher accuracy (96.15%) over FedAvg in all 30 epochs. FedAvg achieved the highest accuracy of 92.30%. FedAvg limits the number of clients to be trained by C, which ranges between 0 and 1. When the data sample is extremely small, there is a possibility that overfitting will occur. When a model is unable to generalise and instead fits

too closely to the training dataset, this phenomenon is known as "overfitting". After 20 and 30 rounds, regardless of the c value, the results remained the same at 92.30% and stopped advancing further after that point as a result of excessive data fitting. The work was done by picking one client from each round of interaction who got a grade of C or higher. The accuracy is higher when the value of C is higher, and data is transmitted between the server and client at the same time as volume increases. However, FPS Optimization converges in less iterations.

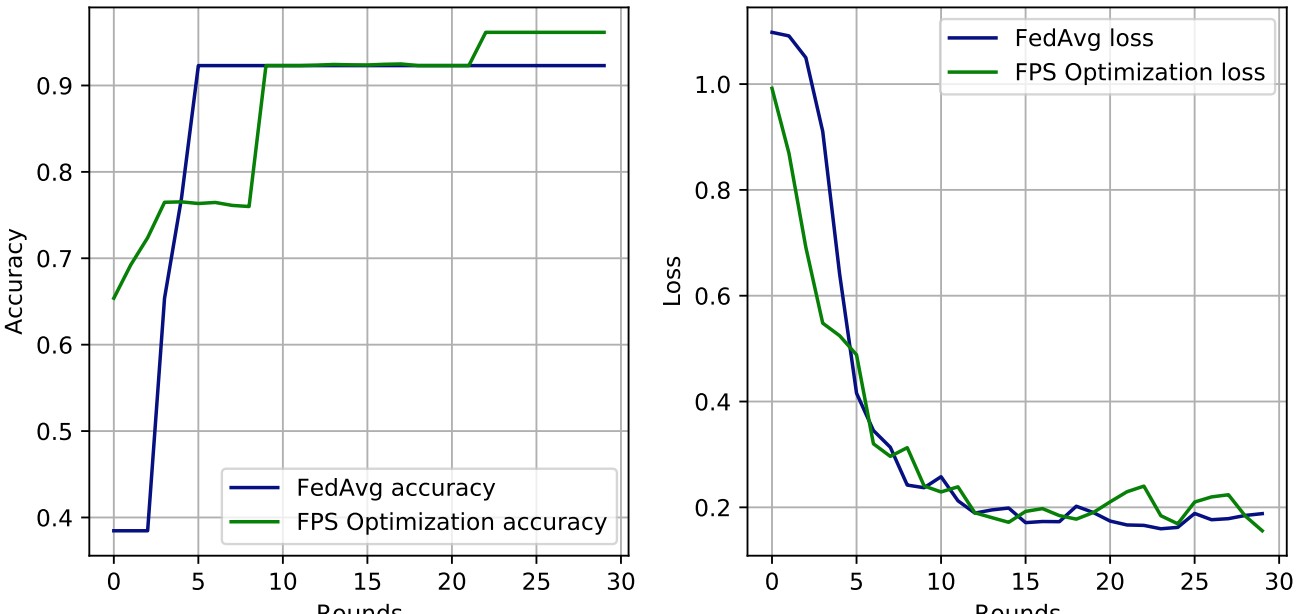

**Figure 6.** Comparison of the FPS Optimization and FedAvg Accuracy and Loss Results for the chest lesion caused by the COVID-19 infection dataset.

**Table 6.** Test Accuracy of the Model.

|  | 10 Rounds | 20 Rounds | 30 Rounds |
|---|---|---|---|
| FPS Optimization | 92.30% | 92.30% | 96.15% |
| FedAvg, C = 1.0 | 78.55% | 92.30% | 92.30% |
| C = 0.5 | 77.54% | 92.30% | 92.30% |
| C = 0.2 | 76.92% | 92.30% | 92.30% |
| C = 0.1 | 65.38% | 92.30% | 92.30% |

*4.6. FPS Optimization Performance for the Chest Lesion Caused by the COVID-19 Infection Dataset*

The objective of this subsection is to assess the model's approach and effectiveness. We train the server model first with samples from the chest lesion caused by the COVID-19 infection dataset. Clients are then assigned a server model according to their client ratio. In our study, we randomly selected 10 clients and assigned them to three client ratios: 0.1, 0.2, and 0.5. For each client device in the dataset, observations are picked at random to make sure the data is correct. Figure 6 illustrates the successful effectiveness of the model in comparison to every iteration.

Since there is no initial model, the global model on the server is trained using the data that is available. Then the model that is initially trained is sent as a reference to all clients by default. The global model is updated with the appropriate client model for each round.

Figure 6 shows the graphs for accuracy and the loss of both FPS optimization and FedAVg for 30 rounds. The accuracy graph shows the results of FPS optimization and

FedAVg for the values of C = 0.1, 0.2, 0.5, and 1.0. When compared to the FedAVg, FPS optimization proved to be more accurate. Similarly, the loss from FPS optimization is less than the loss from FedAVg.

### 4.7. FPS Optimization Performance for the Chest X-ray (Pneumonia) Dataset

The objective of this subsection is to assess the model's approach and effectiveness. We train the server model first with samples from the chest X-ray (pneumonia) dataset. Clients are then assigned a server model according to their client ratio. In our study, we randomly selected 10 clients and assigned them to three client ratios: 0.1, 0.2, and 0.5. For each client device in the dataset, observations are picked at random to make sure the data is correct. Figure 7 illustrates the successful effectiveness of the model in comparison to every iteration.

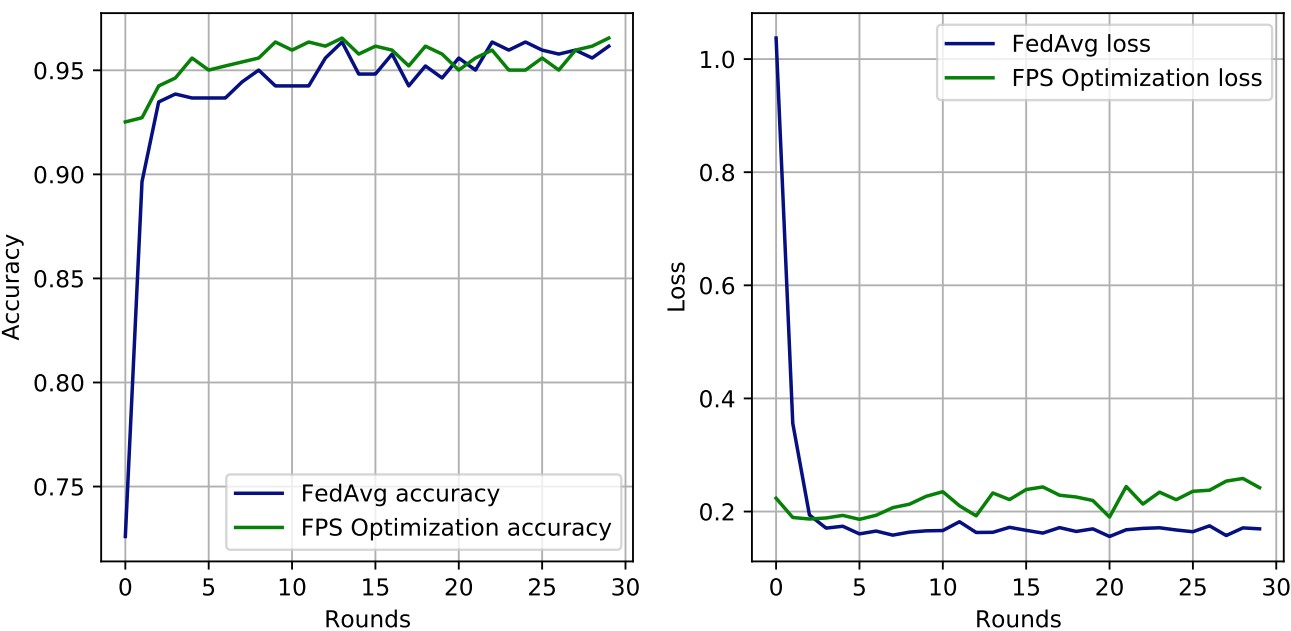

**Figure 7.** Comparison of the FPS Optimization and FedAvg Accuracy and Loss Results for the Chest X-ray (Pneumonia) dataset.

Figure 7 shows the graphs for accuracy and the loss of both FPS Optimization and FedAvg for the pnemonia X-ray dataset. The accuracy graphs show the results of training accuracy and loss for both FPS Optimization and FedAvg. FPS Optimization proved to have a higher accuracy of 96.55% when compared to FedAvg, which showed an accuracy of 96.16%. The Figure 7 also depicts the training-loss results. The FPS Optimization loss is 0.244, and the FedAvg loss is 0.16.

All of the true negatives and false negatives shown in Figure 8 confusion matrix were put into the trained model, which was then used to figure out if non-COVID images were normal or had pneumonia. The result of the true negatives that were transferred in a separate 2 × 2 confusion matrix is shown in Figure 8.

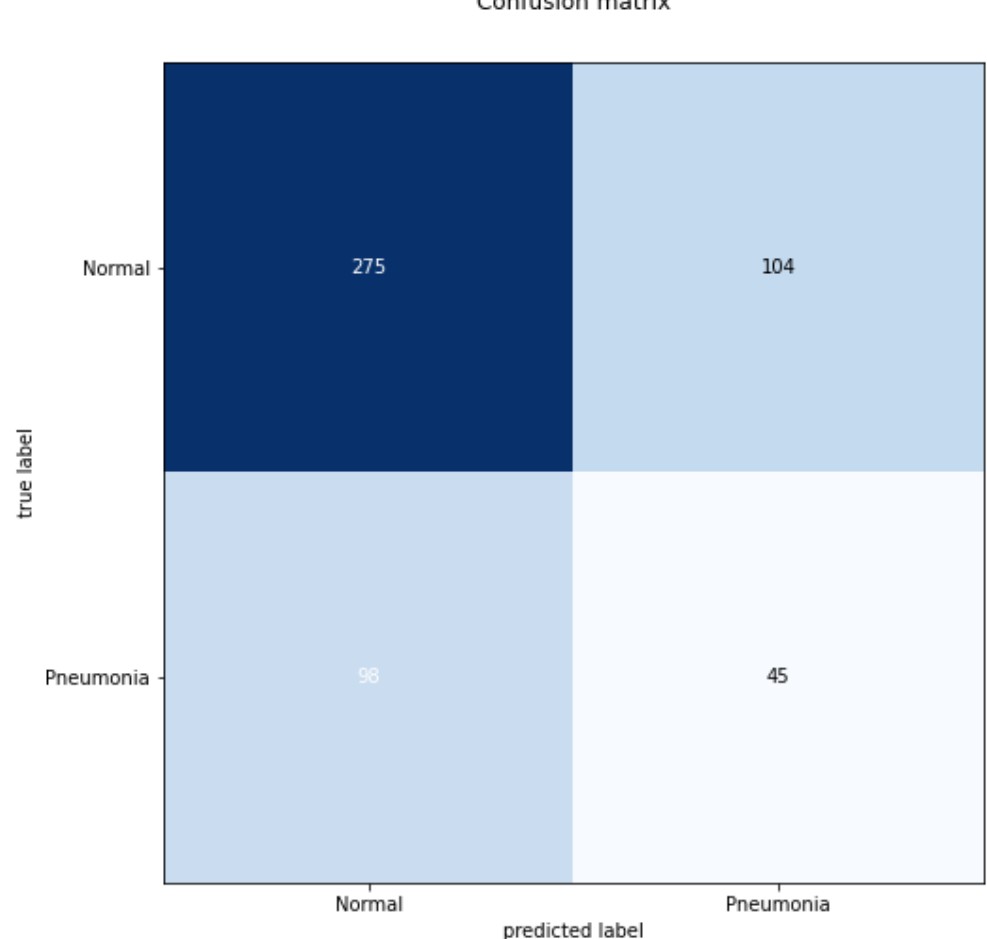

**Figure 8.** Confusion matrix for the Chest X-ray (Pneumonia) dataset.

### 4.8. Analysis and Discussion

The proposed FPS Optimization model achieves improved results using the COVID-19 dataset. Using a computational strategy that optimizes the issue repeatedly by enhancing the candidate solution connected to the model weights. PSO makes it possible to optimize FL by transmitting the optimal weights to the server so that they may be aggregated. In order to assess the effectiveness of the proposed FPS Optimization, we compared the research findings with the most recent state-of-the-art in the line of work. The authors in [49] present a robust framework that takes advantage of advanced machine learning techniques and data analytics techniques for the early detection of Coronavirus diseases using smartphone embedded sensors. The authors in [50] demonstrate how deep learning models can be used to detect COVID-19 by using X-ray images. The authors in [51] proposed a proposed a high-privacy FL system for chest X-ray-based COVID-19 detection. The authors in [52] proposed the Capsule Network-based COVID-CAPS framework for diagnosing COVID-19 based on X-ray images. The comparative analysis of the methodologies is shown in Table 7. With our proposed strategy, FPS Optimization is a big improvement over more traditional FL methods. Our method works better and is more flexible when it comes to refining the hyper-parameters in FL for the COVID-19 dataset. This makes it easier to diagnose the condition. Our technique ensures that the FL performance may be optimized by improving the measures of the client model before it is sent to the server.

**Table 7.** Comparison of the proposed model with recent studies.

| Work | Accuracy | Dataset | Method |
|---|---|---|---|
| Khaloufi et al. [49] | 79% | COVID-19 symptom | ANN is an AI-enabled framework that uses a smartphone to diagnose a chest lesion caused by COVID-19 infection. |
| Horry et al. [50] | 86% | X-ray, ultrasound, CT scan | VGG19 |
| Ho et al. [51] | 95.32% | COVID-19 chest X-ray image Dataset | Differential Privacy Stochastic Gradient Descent (DP-SGD). |
| Afshar et al. [52] | 95.7% | Chest X-ray | COVID-CAPS |
| Proposed work | 96.15%, 96.55% | COVID-19 image dataset, chest X-ray (pneumonia) dataset | FPS Optimization |

## 5. Conclusions and Future Directions

In this article, we apply the FPS optimization method to the prediction of chest lesions caused by the COVID-19 infection dataset and the chest X-ray (pneumonia) dataset in a way that enables clients in various locations to develop a common prediction model without sending the training data to the source. This study came up with a FPS Optimization algorithm that uses PSO to increase the efficiency of the FL and reduce the amount of data that the client sends to the server. The proposed algorithm shares the score value of the model trained on the server for aggregation. The client who has achieved the highest score is responsible for distributing the trained communication of a model to the server. The chest lesion caused by the COVID-19 infection dataset and the chest X-ray (pneumonia) dataset were used to train the suggested method for the first time. We used a three-layer CNN to train the proposed algorithm on the COVID-19 chest lesion dataset and a two-layer CNN to train it on the chest X-ray (pneumonia) dataset. The simulation results showed that the suggested method worked better than other algorithms, with a 96.15% accuracy rate for predictions on the chest lesion caused by the COVID-19 infection dataset and a 96.55% accuracy rate for predictions on the chest X-ray (pneumonia) dataset. PSO makes it possible to optimize FL by transmitting the optimal weights to the server so that they may be aggregated. With our proposed strategy, FPS Optimization is a big improvement over more traditional FL methods. Our method works better and is more flexible when it comes to refining the hyper-parameters in FL for the chest lesion caused by COVID-19 infection dataset.

PSO consumes more calculation time and suffers early convergence at the initial stages. PSO is well suited for smaller datasets when compared to larger datasets. Since the PSO has a poor convergence time and a huge search space, it struggles with larger datasets. Our work can be made more effective by replacing the PSO algorithm with other nature-inspired algorithms.

Future work will include using other methods inspired by nature, such as the Firefly algorithm, the whale optimization algorithm, and artificial bee colony optimization, to make the FL algorithm even better at recognising chest lesions caused by COVID-19 infection and pneumonia diseases. Furthermore, as already discussed, when the CNN layer increases, the size of the strategy increases correspondingly. Hopefully, in the future, we will be able to test different sizes of layers in a model with many layers.

**Author Contributions:** Conceptualization, D.R.K. and T.R.G.; methodology, D.R.K.; software, D.R.K.; validation, D.R.K., and T.R.G.; formal analysis, D.R.K.; investigation, D.R.K.; resources, D.R.K.; data curation, D.R.K.; writing—original draft preparation, D.R.K.; writing—review and editing, D.R.K.; visualization, D.R.K.; supervision, D.R.K.; project administration, D.R.K.; funding acquisition, T.R.G. All authors have read and agreed to the published version of the manuscript.

**Funding:** This research received no external funding.

**Data Availability Statement:** The chest lesion caused by the COVID-19 infection dataset used in this work is taken from the kaggle repository https://www.kaggle.com/datasets/pranavraikokte/covid19-image-dataset (accessed on 5 September 2022).

**Conflicts of Interest:** The authors declare no conflict of interest.

## Abbreviations

The following abbreviations are used in this manuscript:

| | |
|---|---|
| AI | Artificial Intelligence |
| ML | Machine Learning |
| DL | Deep Learning |
| FL | Federated Learning |
| PSO | Particle Swarm Optimization |
| FPS Optimization | Federated Particle Swarm Optimization |
| WHO | World Health Organization |
| CT scans | Computed Tomography scans |
| RT-PCR | Reverse Transcription Polymerase Chain Reaction |
| CXR | Chest X-rays |
| CNN | Convolutional Neural Networks |
| FedAvg | Federated Averaging |
| FedSGD | Federated Stochastic Gradient Descent |
| FedMA | Federated Matched Averaging |
| *pb* | Particle best |
| *gb* | Global best |

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
