# Peer review of "Federated Learning Approach for Early Detection of Chest Lesion Caused by COVID-19 Infection Using Particle Swarm Optimization"

_electronics, doi:10.3390/electronics12030710_

Round 1

Reviewer 1 Report

This research aims to propose an expansion of origin federated learning algorithm with client selection strategy based on Particle Swarm Optimization, which is interested work and seems to be beneficial to distributed learning designing. The overall presentation and writing are good and well-structured.   

Nevertheless, there are some concerns regarding the design of proposed algorithm and the experiment design as well as those results that needed to be addressed before considering for the acceptance.

1. Detecting COVID-19 on X-ray image is controversial problems: Covid-19 is only a kind of Pneumonia thus there is not any of single feature of COVID-19 on chest radiography is specific or diagnostic for the precise detection. It is infeasible (if not impossible) to differentiate between COVID-19 versus Pneumonia solely on X-ray image. Hence, while establishing the direction of this research is about COVID-19 detection sounds impressive, it’s really lacking scientific foundation. The research might be more applicable if it changes it’s direction to detect chest diseases (Atelectasis, Edema, etc.) or chest lesion caused by COVID-19 infection (Consolidation, Ground Glass Opacity, etc.).

2. To efficiently delivering the ideas, concepts about the proposed algorithm, authors should provide simple, brevity description about what is their proposal before going to detail (Section 3). For instance, at the end of section 2.3, after summarized related works, there should be a comparison between the previous works with the proposed algorithm such as “this algorithm not aggregating the client model’s weights, instead the client model with highest score would be re-broadcasted to all the clients to incremental learn from the local data”…etc

3. The explanation about the proposed algorithm (line 189 - 198) needs to be revised such that more details but by the general expression (the expression by notation and math proof style is fine but too ambiguous and hard to understand). For example, the author should clarify that by their proposed algorithm, “the global model is partially incorporated to client model by the PSO layer by layer (line 13-14 in Algorithm 2 pseudo code) and then the client models are reinforced by their local data before being judged by pb evaluation”…etc

4. The proposed algorithm are designed in the way that only the highest score client model is sent back to the server in order to be broadcasted to other clients in the new federated learning round. This might be the limitation as in the initialized / beginning, the global model needs to be trained first by the server (line 255-256) otherwise the clients will incorporate dump global model (it means the global model did not carrying any knowledge) thus may potentially degrade the whole process. But in practical, the server in federated learning does not hold any data thus asking server to train the global model first is really a disadvantage. The author should justify this disadvantage of their proposed.

5. The author did not describe how is the data are distributed to clients to emulate the federated learning such as how many image a client held, are client’s data IID (Independent and identically distributed)…etc. The author did not specifically mention on which metric the pb represented for in their experiment (even they declared that can be the loss or accuracy value, line 170, and it seems that they implied that the lower pb the better, line 192, means it must be the loss value). These concerns should be addressed.

6. It might be worth to analyze the role of PSO’s parameters which are inertia alpha, c1, c2 as well as rand1 and rand2 functions. How would those parameters control the performance of the proposed algorithm and what are the default values the author employed in their experiment? These settings should be declared in the experiment section for the reproducibility.

7. The experiment is conduct with the input image size is only 32x32, means the X-ray images were scaled-down too small that lost all the information and what the CNN might learn is only the properties/characteristics of the sources of images or the image itself (brightness, saturation, mark-up…) but not those real disease-relevant features of the COVID-19. It’s highly recommended that the author should re-conduct the experiment with higher resolution for neutral result in term of accuracy of COVID-19 detection, and higher number of clients (50-100) to obtain the compelled results that can justify for the effectiveness of the proposed algorithm. Moreover, for transparency judgment, the author should additionally supply the confusion matrix and additionally employ the ROC or F1-score as evaluation metric rather than the accuracy only.

8. Finally, the dataset employed in the experiment is too small in volume thus the improvement of proposed algorithm compares to classic FedAvg (3.85%, line 279) might be gained by chance and is not statistically significant. It’s also highly recommended that the author should re-conduct their experiment with another bigger chest X-ray dataset from RSNA, NIH (the TorchXRay Vision is library for chest X-ray datasets and models which is highly suggested) for transparent evaluation.

Author Response

  1. Detecting COVID-19 on X-ray image is controversial problems: Covid-19 is only a kind of Pneumonia thus there is not any of single feature of COVID-19 on chest radiography is specific or diagnostic for the precise detection. It is infeasible (if not impossible) to differentiate between COVID-19 versus Pneumonia solely on X-ray image. Hence, while establishing the direction of this research is about COVID-19 detection sounds impressive, it’s really lacking scientific foundation. The research might be more applicable if it changes it’s direction to detect chest diseases (Atelectasis, Edema, etc.) or chest lesion caused by COVID-19 infection (Consolidation, Ground Glass Opacity, etc.).

Statement: As per the above mentioned query, the term COVID-19 is changed to chest lesion caused by the COVID-19 infection in the revised manuscript.

  1. To efficiently delivering the ideas, concepts about the proposed algorithm, authors should provide simple, brevity description about what is their proposal before going to detail (Section 3). For instance, at the end of section 2.3, after summarized related works, there should be a comparison between the previous works with the proposed algorithm such as “this algorithm not aggregating the client model’s weights, instead the client model with highest score would be re-broadcasted to all the clients to incremental learn from the local data”…etc

Statement: As per the above mentioned query, the related works and comparisons between the previous works are updated in the revised manuscript. Please refer to the section 2.3.

  1. The explanation about the proposed algorithm (line 189 - 198) needs to be revised such that more details but by the general expression (the expression by notation and math proof style is fine but too ambiguous and hard to understand). For example, the author should clarify that by their proposed algorithm, “the global model is partially incorporated to client model by the PSO layer by layer (line 13-14 in Algorithm 2 pseudo code) and then the client models are reinforced by their local data before being judged by pb evaluation”…etc

Statement: As per the above mentioned query, Algorithm 2 pseudocode is explained layer by layer in the revised manuscript. Please refer to the section 3.

  1. The proposed algorithm are designed in the way that only the highest score client model is sent back to the server in order to be broadcasted to other clients in the new federated learning round. This might be the limitation as in the initialized / beginning, the global model needs to be trained first by the server (line 255-256) otherwise the clients will incorporate dump global model (it means the global model did not carrying any knowledge) thus may potentially degrade the whole process. But in practical, the server in federated learning does not hold any data thus asking server to train the global model first is really a disadvantage. The author should justify this disadvantage of their proposed.

Statement: As per the above mentioned query, the proposed algorithm is updated in the revised manuscript. Please refer to the section 3.

  1. The author did not describe how is the data are distributed to clients to emulate the federated learning such as how many image a client held, are client’s data IID (Independent and identically distributed)…etc. The author did not specifically mention on which metric the pb represented for in their experiment (even they declared that can be the loss or accuracy value, line 170, and it seems that they implied that the lower pb the better, line 192, means it must be the loss value). These concerns should be addressed.

Statement: As per the above mentioned query, dataset partitioning is added to the revised manuscript. Please refer to the section 4.3.

  1. It might be worth to analyze the role of PSO’s parameters which are inertia alpha, c1, c2 as well as rand1 and rand2 functions. How would those parameters control the performance of the proposed algorithm and what are the default values the author employed in their experiment? These settings should be declared in the experiment section for the reproducibility.

Statement: As per the above mentioned query, we have clearly mentioned inertia alpha, c1, c2, as well as rand1 and rand2 functions in the revised manuscript. Please refer to the section 2.2.

  1. The experiment is conduct with the input image size is only 32x32, means the X-ray images were scaled-down too small that lost all the information and what the CNN might learn is only the properties/characteristics of the sources of images or the image itself (brightness, saturation, mark-up…) but not those real disease-relevant features of the COVID-19. It’s highly recommended that the author should re-conduct the experiment with higher resolution for neutral result in term of accuracy of COVID-19 detection, and higher number of clients (50-100) to obtain the compelled results that can justify for the effectiveness of the proposed algorithm. Moreover, for transparency judgment, the author should additionally supply the confusion matrix and additionally employ the ROC or F1-score as evaluation metric rather than the accuracy only.

Statement: As per the above mentioned query, the experiment is conducted with the input image size 150 x 150. We have re-conducted the experiment with other dataset, and the results are updated in the revised manuscript. Please refer to the section 4.7.

  1. Finally, the dataset employed in the experiment is too small in volume thus the improvement of proposed algorithm compares to classic FedAvg (3.85%, line 279) might be gained by chance and is not statistically significant. It’s also highly recommended that the author should re-conduct their experiment with another bigger chest X-ray dataset from RSNA, NIH (the TorchXRay Vision is library for chest X-ray datasets and models which is highly suggested) for transparent evaluation.

Statement: As per the above mentioned query, we have re-conducted the experiment with another bigger chest X-ray dataset in the revised manuscript.

Reviewer 2 Report

Dear Authors,

   Happy New Year 2023! Many thanks for your manuscript submission to MDPI Journal of Electronics. This paper applies particle swarm optimization (PSO) approach and federated learning scheme, which jointly work for early detection of COVID-19 before its outbreak. This short paper has some useful validations and the results of predicted accuracy is quite high, the outcome is the claim to develop a "novel" framework on early detection of COVID-19. 

   I think this paper needs some revision to fix the technical problems as well as formatting issues in order to qualify entering the double decision process, where the problematic issues can be summarized as follows:

   a) Abstract: in Line 18, the claim on "novel framework" needs attention. I think federated learning and PSO based schemes are common approaches, jointly apply both of them are just progressive applicable instead of "novel".

   b) Introduction: in Lines 22-39, it is quite too simple to narrate the COVID-19 related backgrounds, please consider applying some rewrite on the opening paragraph. Besides, in Lines 65-81, the "procedure of proposed methodology", can be expanded to a summary of major contributions on your set of work, and better to preserve the 3 manifolds with a little bit more specific details. Besides, in Line 85 at Page 2, section 5 --> Section 5.

   c) Background and Related Work: in Lines 120-126, since PSO is a classical scheme, you may just cite it instead of showing a simple mathematical model, meanwhile, in Lines 137-154, I think the authors need to take some concern on the improved PSO scheme and more recently derived methods; the current section looks a bit too generic. 

   d) Proposed methodology: Above Line 184, this section shows a formula on "FPS optimization weights", while the expected evaluation metrics are missing, and the basic idea of simple CNN, simple PSO, are suggested to be improved. 

   e) Figures and Tables: the titles of each figure lacks specific details, i.e., Figs. 1-3. The middle-alignments are preferred for Tables 1-5. Also, I think that the tabulated results also seems a bit too simple, do you have alternative plans to fix that potential problem? Thanks a lot!

   f) Experimental Results: In Line 243, Accuracy should be defined. Right above Line 262, the results of 20 rounds in Table 5 are all 92.30%, can you explain why, or apply other number rounds to clarify this issue? Besides, in Lines 244-250, I think the outcomes of Fig. 5 and Table 5, are suggested to be explained separately, which could be clearer. 

   g) Analysis and Discussion: in Lines 260-268, if better results are included, this part can be expanded to a larger, independent section. 

   h) Conclusions and Future work:The last paragraph looks a bit generic. The updated version, needs to be an extension on discussion your work and any useful investigations. Besides, there must be a paragraph on future work to fill in any potential defects, which may consist of the challenging topics related to your work, opening questions and possible solutions, some future orientations of subsequent studies, etc. Please improve the structure and optimize the organization of your conclusion section. Thanks a lot! 

   i) References: The current version has some obvious problems, which MUST be fixed. 1) Incomplete citation on some journals, i.e., [18], [21] and [22]. 2) Abbreviated styles should be applied to each journal citations, and capitalize the first letter of each word. 3) Please make up the missed information for conference proceedings, i.e., [8], [10], [19], and [29]. 4) Supplement a few more closely related publications to COVID-19 detection by deep learning and federated testbed based approachs in Years 2020-2023. 

   Minor problematic issues suggested to be calibrated in your revisions: 

   a) The literal quality of English should be further improved. I recommend the authors prepare one round of peer-review again on this paper, which may include proofreading and grammatical checks in your updates.

   b) The font size, italic style of characters in the figures (i.e., Fig. 2 and Fig. 4), should be uniform. Please adjust the required points in your updated version.

   c) Please fix the related issues on linespacing, image quality, location and positioning of figures and tables. Be careful and strictly be consistent with the required styles as specified by MDPI.  

   We look forward to seeing your updated research article coming into further acceptance. Good luck!

With warm regards,

Yours sincerely,

Author Response

  1. a) Abstract: in Line 18, the claim on "novel framework" needs attention. I think federated learning and PSO based schemes are common approaches, jointly apply both of them are just progressive applicable instead of "novel".

Statement:  As per the above mentioned query, descriptions have been provided in the revised manuscript. Please refer to abstract.

  1. b) Introduction: in Lines 22-39, it is quite too simple to narrate the COVID-19 related backgrounds, please consider applying some rewrite on the opening paragraph. Besides, in Lines 65-81, the "procedure of proposed methodology", can be expanded to a summary of major contributions on your set of work, and better to preserve the 3 manifolds with a little bit more specific details. Besides, in Line 85 at Page 2, section 5 --> Section 5.

Statement:  As per the above mentioned query, descriptions have been provided. section 5 is changed to Section 5 in the revised manuscript. Please refer to Introduction.

  1. c) Background and Related Work: in Lines 120-126, since PSO is a classical scheme, you may just cite it instead of showing a simple mathematical model, meanwhile, in Lines 137-154, I think the authors need to take some concern on the improved PSO scheme and more recently derived methods; the current section looks a bit too generic. 

Statement: As per the above mentioned query, descriptions have been provided in the revised manuscript. Please refer to Background and Related Work.

  1. d) Proposed methodology: Above Line 184, this section shows a formula on "FPS optimization weights", while the expected evaluation metrics are missing, and the basic idea of simple CNN, simple PSO, are suggested to be improved. 

Statement: As per the above mentioned query, descriptions have been provided in the revised manuscript. Please refer to section 3.

  1. e) Figures and Tables: the titles of each figure lacks specific details, i.e., Figs. 1-3. The middle-alignments are preferred for Tables 1-5. Also, I think that the tabulated results also seems a bit too simple, do you have alternative plans to fix that potential problem? Thanks a lot!

Statement: As per the above mentioned query, all the figures and tables are updated in the revised manuscript.

  1. f) Experimental Results: In Line 243, Accuracy should be defined. Right above Line 262, the results of 20 rounds in Table 5 are all 92.30%, can you explain why, or apply other number rounds to clarify this issue? Besides, in Lines 244-250, I think the outcomes of Fig. 5 and Table 5, are suggested to be explained separately, which could be clearer. 

Statement: As per the above mentioned query, experimental results are updated in the revised manuscript. The outcomes of Fig. 6 and Table 6 are explained separately in the revised manuscript.

  1. g) Analysis and Discussion: in Lines 260-268, if better results are included, this part can be expanded to a larger, independent section.

Statement: As per the above mentioned query, analysis and discussion are expanded in the revised manuscript.

  1. h) Conclusions and Future work:The last paragraph looks a bit generic. The updated version, needs to be an extension on discussion your work and any useful investigations. Besides, there must be a paragraph on future work to fill in any potential defects, which may consist of the challenging topics related to your work, opening questions and possible solutions, some future orientations of subsequent studies, etc. Please improve the structure and optimize the organization of your conclusion section. Thanks a lot! 

Statement: As per the above mentioned query, conclusions and future work descriptions have been provided in the revised manuscript.

  1. i) References: The current version has some obvious problems, which MUST be fixed. 1) Incomplete citation on some journals, i.e., [18], [21] and [22]. 2) Abbreviated styles should be applied to each journal citations, and capitalize the first letter of each word. 3) Please make up the missed information for conference proceedings, i.e., [8], [10], [19], and [29]. 4) Supplement a few more closely related publications to COVID-19 detection by deep learning and federated testbed based approachs in Years 2020-2023. 

Statement: As per the above mentioned query, all the incomplete citations are updated, and references are included in the revised manuscript.

Minor problematic issues: 

  1. a) The literal quality of English should be further improved. I recommend the authors prepare one round of peer-review again on this paper, which may include proofreading and grammatical checks in your updates.

Statement: As per the above mentioned query, proofreading has been done on the revised manuscript.

  1. b) The font size, italic style of characters in the figures (i.e., Fig. 2 and Fig. 4), should be uniform. Please adjust the required points in your updated version.

Statement: As per the above mentioned query, the font size and italic style of characters in the figures are updated in the revised manuscript.

  1. c) Please fix the related issues on linespacing, image quality, location and positioning of figures and tables. Be careful and strictly be consistent with the required styles as specified by MDPI.  

Statement: As per the above mentioned query, we fixed all the related issues with figures and tables in the revised manuscript.